# Individualized Prognostic Prediction of the Long-Term Functional Trajectory in Pediatric Acquired Brain Injury

**DOI:** 10.3390/jpm11070675

**Published:** 2021-07-18

**Authors:** Erika Molteni, Marta Bianca Maria Ranzini, Elena Beretta, Marc Modat, Sandra Strazzer

**Affiliations:** 1School of Biomedical Engineering & Imaging Sciences, King’s College London, London SE1 7EU, UK; erika.molteni@kcl.ac.uk (E.M.); marta.ranzini@kcl.ac.uk (M.B.M.R.); marc.modat@kcl.ac.uk (M.M.); 2Acquired Brain Injury Unit, Scientific Institute IRCCS E. Medea, 22040 Bosisio Parini, Italy; sandra.strazzer@lanostrafamiglia.it

**Keywords:** single-subject recovery prediction, trajectory prediction, Functional Independence Measure for children (WeeFIM), acquired brain injury (ABI), mixed models, structural equation modelling (SEM), latent class analysis

## Abstract

In pediatric acquired brain injury, heterogeneity of functional response to specific rehabilitation treatments is a key confound to medical decisions and outcome prediction. We aimed to identify patient subgroups sharing comparable trajectories, and to implement a method for the early prediction of the long-term recovery course from clinical condition at first discharge. 600 consecutive patients with acquired brain injury (7.4 years ± 5.2; 367 males; median GCS = 6) entered a standardized rehabilitation program. Functional Independent Measure scores were measured yearly, until year 7. We classified the functional trajectories in clusters, through a latent class model. We performed single-subject prediction of trajectory membership in cases unseen during model fitting. Four trajectory types were identified (post.prob. > 0.95): high-start fast (*N* = 92), low-start fast (*N* = 168), slow (*N* = 130) and non-responders (*N* = 210). Fast responders were older (chigh = 1.8; clow = 1.1) than non-responders and suffered shorter coma (chigh = −14.7; clow = −4.3). High-start fast-responders had shorter length of stay (c = −1.6), and slow responders had lower incidence of epilepsy (c = −1.4), than non-responders (*p* < 0.001). Single-subject trajectory could be predicted with high accuracy at first discharge (accuracy = 0.80). In conclusion, we stratified patients based on the evolution of their response to a specific treatment program. Data at first discharge predicted the response over 7 years. This method enables early detection of the slow responders, who show poor post-acute functional gains, but achieve recovery comparable to fast responders by year 7. Further external validation in other rehabilitation programs is warranted.

## 1. Introduction

Acquired brain injury (ABI) has been recently conceptualized as a chronic disease [1] with cognitive and motor sequelae. In the most severe cases, functional changes after brain injury may occur over a whole decade [2].

After a brain injury, 70% of children receiving motor rehabilitation improve their motor functions within 1 year from injury [3]. However, the prime difficulty encountered by professionals in the inpatient rehabilitation settings is the early prediction of the patients’ treatment outcome. Indeed, considerably different responses are observed to standardized treatment programs, in individuals with seemingly comparable clinical conditions at post-acute admission, and who receive equivalent intensity and elements of care [4]. Response is generally assessed through scales, of which the Functional Independence Measure (FIM) [5] is the one recommended and most frequently used [6].

FIM was employed in public health, to determine the prevalence and nature of residual disability after inpatient rehabilitation for children with traumatic injuries [7]. In a recent systematic review studying children with ABI [8], the pediatric version of the FIM scale (WeeFIM) resulted to be the elective tool for functional assessment, being more suitable for the inpatient setting, and quicker to administer, compared to the Pediatric Evaluation of Disability Inventory (PEDI). It also showed less ceiling effects. WeeFIM was also employed in pediatric cohorts with ABI as a measure of global functional outcome at admission [9] and discharge from hospital [10,11,12], and as a robust clinical indicator of motor function correlating with neuroimaging biomarkers, such as the DTI measures in the corticospinal tract [13,14] and the volume of putamen [15]. However, although FIM can track functional improvements reliably over time, there is no score cut-off to predict subsequent recovery trajectory or outcome.

In adults, typological analysis [2,16], revealed either three or four distinctive long-term motor and cognitive patterns in traumatic brain injury (TBI). More than half of adults achieve near-maximal recovery by 1-year post-injury, while functional improvements or declines beyond year 5 are debated [2].

Pediatric research has confirmed the increased risk of poor outcome for higher injury severity, longer unconsciousness [17,18], longer time to follow commands [19], and anoxic etiology [18,20,21,22], but no effect of gender on outcome [18,22]. Additionally, age under 6 years predicted negative long-term outcome in anoxic [18], but not in TBI [19]; and older children were found to make faster gross motor recovery [22]. Typological analyses of functional trajectories in pediatrics are virtuous exceptions [19,22], rather than a consolidated field of study. Similarly to observations in adults, pediatric studies described subgroups of patients who did not show any recovery trend. Limitations were short timeframe of observation [19] and employment of functional assessments other than FIM, such as the Gross Motor Function Measure-66 [22].

In this study, we modelled the functional trajectories of 600 pediatric patients (0–18 years) with ABI of both traumatic and non-traumatic origins, as measured by yearly FIM scores.

The objectives of the work are:

(a) to apply a completely data-driven approach to identify subgroups of patients sharing comparable trajectories, whose characteristics could inform the rehabilitation practice.

(b) to design a framework for single-subject prediction of trajectories, based on the clinical and functional conditions at first discharge.

## 2. Materials and Methods

### 2.1. Participants, Inclusion Criteria, Measures at Admission, Discharge and Outcome

From a candidate set of 926 eligible pediatric patients with ABI admitted to the Scientific Institute IRCCS E. Medea, Bosisio Parini, Italy between 2003 and 2018, we included 600 subjects who met the inclusion criteria: (1) diagnosis of ABI, (2) age < 18 years at ABI event, (3) complete medical records of the acute phase, and (4) admission to the rehabilitation center between January 2003 and December 2018, after transfer from intensive care or neurosurgery units of other hospitals. Exclusion criteria were: (1) documented evidence of a previous ABI of traumatic or non-traumatic (i.e., anoxic, vascular or infectious) etiology, (2) previous diagnosis of any congenital (including genetic) disease or syndrome, (3) previous known cognitive and/or motor disability, according to caregivers’ interview (Appendix A).

Demographic and clinical information, including the neurological condition and disorders of consciousness [23,24], were collected at admission and discharge of the initial in-stay, and were available for all patients (Table 1, and Appendix A). At admission, they included the days of coma, Glasgow Coma Score (GCS) at event [25], Glasgow Outcome Score (GOS) [26], and Disability Rating Scale (DRS) [27]. At discharge, the GOS, DRS, and information about recovery were collected again. Data at yearly follow-ups, until the 7th year after first admission, were collected by assessors blinded to the first patients’ in-stay during short hospital visits (1–3 days), and were included in this study when available (Table 2). The rehabilitation program consisted in a set of standardized elements (exercises and activities), organized as an in-stay intensive treatment [28], followed by a home-care program (Appendix A). It is of note that the rehabilitation elements were tailored to patients, but neither intensity was ever decreased with respect to the standard, nor therapy withdrawn based on poor recovery. The research was approved by the Institutional Review Board of IRCCS E. Medea (N. 58/19—CE) and all caregivers consented to data collection. Children verbally consented when allowed by age and severity of disease. Research was conducted in compliance with the Ethical Standards for Research with Children (https://www.srcd.org/about-us/ethical-standards-research-children, accessed on 17 July 2021). A subsample of this cohort was previously investigated in [28]. The complete dataset is available at https://zenodo.org, accessed on 17 July 2021, doi:10.5281/zenodo.4153962, uploaded on 29 October 2020.

### 2.2. Functional Independence Measure (FIM) and Functional Independence Measure for Children (WeeFIM)

The primary outcome of interest is the patients’ functional status. This was assessed through the FIM, or WeeFIM [29,30] when appropriate to age. WeeFIM, a pediatric normalization of FIM, has been validated in children with normal development [30], developmental disabilities [31,32], and ABI [32,33]; and it was employed in the evaluation of trauma-related disability at short- and long-term [10,19,34]. WeeFIM/FIM is composed of 3 subscales rating self-care, mobility and cognitive abilities (see Appendix A for details). Patients were evaluated with the WeeFIM/FIM by certified staff at first admission (within 3 days), discharge, and at each follow-up until 7 years after first admission.

### 2.3. Statistical Analysis

After applying descriptive statistics to the dataset, preliminary Spearman’s correlation analysis between demographic and clinical variables identified independent information. Selected covariates (r < 0.5; alpha = 0.01 Bonferroni corrected) were: gender, age at event, etiology, age at admission, length of stay (LOS), days of coma, need of decompressive craniotomy/neurosurgery (DC/N), and insurgence of epilepsy during in-stay. Data at first admission and discharge were available from all patients. Missing data at follow-ups are depicted in Table 2. Covariate data are complete. For descriptive data analysis, we designed a parsimonious conditional model, which combined all significant covariate-time effect associations (Appendix A), and we conducted a survival analysis to investigate the characteristics of patients who remained engaged with the rehabilitation service, as well as to assess the risk to drop out (Appendix A).

#### 2.3.1. Clustering

Aggregation of similar recovery trajectories was designed to identify responders to treatment and non-responders. Generalized structural equation model (SEM) estimation with one discrete latent class was applied: one equation predicted the class from the covariates, one explained the outcome trajectory, and one incorporated the survival in the rehabilitation service over time (Appendix A). No constraint was imposed on the selection and ordering of the classes, which were checked a posteriori. To maximize the use of available data, subjects with missing follow-ups were included in the analysis, but with no imputation of missing time points. Clustering was conducted for 2, 3, 4, and 5 classes. The role of covariates in the latent class formation was examined against class 1 by default. Bayesian Information Criterion was calculated, as well as the marginal and posterior probabilities, for each class.

Model validation then assessed the clustering stability. The database was randomized into two sets; constrained and unconstrained multiple-group generalized SEMs were run. Likelihood ratio tests for differences were calculated for 2, 3, 4 and 5-classes clustering, which were non-significant in all cases, indicating stability.

To obtain a final unbiased estimate of class membership for each subject, we then performed the clustering twice in cross design: one subset was used to train the model and the other to estimate the class (unseen cases), and vice versa. STATA v.16.1 software was used.

#### 2.3.2. Prediction

Single-subject prediction of the long-term trajectory membership was performed, using demographic, clinical and functional data available by the day of first discharge. The days from event to discharge were used in place of the days of coma, if shorter. Only FIM data at first admission and discharge were included for prediction. A logistic regression model was chosen. Aforementioned clustering results on unseen cases were taken as “ground truth” for prediction. Each of the two randomized sets (halves) were further divided into training and test sets, in k consecutive folds. Each fold was used once as a validation set of unseen cases, and in k −1 remaining iterations to form the training set. The model parameters were retrained at each iteration, with no transfer, and predictions were drawn for each (unseen) validation set. The procedure was repeated for both randomized sets. After binarizing the classifier output according to maximal probability, Receiver Operating Characteristic (ROC) curves were calculated for each class. Prediction performance was calculated on the whole dataset, by joining results from the two randomized sets, and expressed through accuracy, precision, and Cohen kappa score. Confusion matrices were calculated for providing details on the multi-class classification errors. Scikit-learn toolbox (v.18.1, https://scikit-learn.org, accessed on 17 July 2021), based on Python language, was used for implementation. See Appendix A for a graphical depiction of the method workflow.

## 3. Results

### 3.1. Sample Description

We studied 600 children and adolescents (61.2% males) with a history of acquired brain injury and aged 0–18 years at the time of injury. The youngest was 3 months old. All patients were in the post-acute stage, and they were admitted for rehabilitation 43 days after the ABI event as median time (IQR 27; 69). Their median LOS during the first admission at our rehabilitation center was 112 days (IQR 69; 163), which was not significantly different between the traumatic and non-traumatic cases. The clinical characteristics of the whole sample, traumatic and non-traumatic sub-groups, are reported in Table 1.

All patients had complete FIM assessments at first admission and discharge (Appendix A). Available data at the following yearly time points are reported in Table 2. One year after admission, 121 (20.2%) patients still scored at the bottom of the FIM scale, showing no improvement.

Among the 326 non-eligible children, causes of exclusion were previous disability (*N* = 54), congenital disease (*N* = 255), and previous ABI (*N* = 17).

### 3.2. Trajectory Descriptors and Bias over Time

Descriptive analysis with mixed models showed that older age and shorter coma related to initial higher scores at total FIM and at all domains at admission. Longer LOS related to initial lower scores, and to reduced curvature at total FIM. Anoxic etiology presented with slower progress than TBI course. DC/N and presence of epilepsy were associated with slower tangent at admission, for total FIM and the selfcare domain. Epilepsy also related to slower tangent in the mobility domain (complete results in Appendix A). Survival analysis indicated that patients who remained in the rehabilitation service for longer were gradually less likely to drop out. Patients with epilepsy (Chi-sq = 1692, *p* << 0.0001) and those who received DC/N (Chi-sq = 1688, *p* << 0.0001) were more likely to remain in the service for longer (Appendix A).

### 3.3. Clustering

Data-driven clustering into two classes showed very high posterior probability (0.99 for both classes), indicating very good classification (Figure 1). It overall distinguished between clusters that we named responders (*N* = 295) and non-responders (*N* = 305) to treatment. Non-responders, who showed overall a small increase in total FIM values over time, differed in significantly longer period of coma (c = 5.9), higher incidence of anoxic etiology (c = 1.9) and younger age (c = −1.3), in comparison to responders, who had marked total FIM increase over time. Of note, clusters were obtained computationally (i.e., in data-driven manner) and named after the observed overall trajectory; thus, the non-responders cluster includes both patients who show no and (very) limited clinical improvement over time.

Clustering into three classes maintained very high posterior probability (≥0.95). It enabled the distinction between what we called fast-responders (*N* = 195), who had quicker FIM increase, and slow-responders (*N* = 180) to treatment, in addition to the previously identified non-responders (*N* = 225). Fast-responders additionally differed for significantly shorter LOS in the rehabilitation center (c = −0.92), in comparison to non-responders. Slow responders had lower incidence of epilepsy (c = −1.3), in comparison to non-responders. Both classes of responders had fewer days of coma (cfast = −12.2; cslow = −4.3), older age (cfast = 2.2; cslow = 0.93) and less incidence of anoxic cases (cfast = −2.9; cslow = −2.3) as a group.

Clustering into four classes maintained very high posterior probability (≥0.95), also enabling further identification of two sub-classes of fast responders: high-start fast responders (*N* = 92), who already had overall high FIM scores at admission, and low-start fast responders (*N* = 168), in addition to slow responders (*N* = 130) and non-responders (*N* = 210). Both high-start and low-start fast responders were older (chigh = 1.8; clow = 1.1) than non-responders and suffered a shorter period of coma (chigh = −14.7; clow = −4.3). High-start fast-responders had shorter LOS (c = −1.6), and slow responders had lower incidence of epilepsy (c = −1.4), in comparison to non-responders, consistently with results from three-class clustering (*p* < 0.001 for all reported results).

Subject clustering based on each FIM domain (selfcare, mobility and cognition) provided comparable results (not shown).

Clustering into five classes was attempted, but a drop in posterior probability was observed for three out of five classes (≥0.92), thus implying less robust classification.

It is of note that, over time, 61/600 individuals, identified as slow responders, opted out of the service.

### 3.4. Single-Subject Prediction at First Discharge

Binary prediction (two classes) of total FIM trajectories correctly classified unseen responders and non-responders with accuracy of 0.93 (Figure 2). The feature most contributing to prediction was the difference in total FIM value between first admission and discharge, followed by the total FIM value at first admission, the days of coma, anoxic etiology and age, respectively. This agrees with the features most relevant to the trajectory clustering. Prediction among three classes correctly classified fast-, slow- and non-responders with an accuracy of 0.84, and among four classes (high-start fast-, low-start fast-, slow- and non-responders) with an accuracy of 0.80. The features most relevant to the trajectory clustering remained unchanged overall, with the only exception of epilepsy, which gained importance in the discrimination between the four classes. Prediction errors were committed between neighboring trajectories only. Figure 3 shows the prediction performances among four classes, based on data in the FIM domains. The worst prediction was obtained for slow-responders and when relying on scores in the selfcare and cognitive domain. Effect sizes are reported in Appendix A

## 4. Discussion

This longitudinal cohort study monitored the functional recovery of 600 pediatric patients with ABI receiving standardized rehabilitation, over a time span of 7 years, and with yearly follow-up assessments. The study identified four distinctive patterns of recovery time-course; and it tested the feasibility of single-subject trajectory prediction, based on the information available at first discharge.

Using a data-driven approach, we identified four subgroups of individuals who followed distinct trajectories of recovery over time, between first admission and year 7. Fifteen percent were high-start fast responders, having already achieved near-maximal recovery 1 year post-injury, 28% were low-start fast responders, 22% slow responders, and 35% non-responders. Regardless of whether clustering was conducted on total FIM scores or on scores at specific FIM domains (selfcare, mobility or cognition), subjects grouped consistently, and produced similar group trajectories. Clinically, this implies that the assignment of a patient to their group remains fairly independent from the specific clustering method, and it is stable across the FIM domains.

The selection of four recovery patterns as an optimal number agrees with work on adults [2,16]; however, the statistics differs, with more than half of the adult sample classified in a top-of-scale high-functioning group [2]. Admittedly, adult research was affected by bias towards cases with good prognosis, while the cohort presented here has bias towards complications (epilepsy, need of neurosurgery) over time.

In our study, each of the four patterns showed association with varied demographic and clinical characteristics. Clustering analysis confirmed that longer coma and LOS corresponded to lower chances of favorable trajectory of recovery. In adults, Hammond et al. [35] found a class analogous to the non-responders, grouping those with severest injuries and the longest LOS. Lu et al. [16] found that longer in-hospital care (ICU and rehabilitation unit stays) was significantly related to worse recovery trajectory membership. We identified a group of slow responders, who failed to reach good functional level by year 1, but continuously improved until approaching very good (>80) functional recovery by year 7. In adults, Hammond et al. [2] identified a comparable group. This evidence advocates for careful evaluation of those patients who show poor improvement by the time of first discharge, as many cases conceal great long-term potential if treated with perseverance. Slow responders should be regarded as patients with continuing needs of assistance. Takeover by local healthcare services is essential in these cases, to deliver transitional care at home or in community-based groups, and to support the continued improvement during the chronic phase, when therapy standardization and monitoring are most difficult. In this study, 61/600, identified as slow responders, opted out of the service at some point during recovery. Their outcome at 7 years is unknown and cannot provide a comparison.

The rate of recovery is known to be an important prognostic variable in adults [36,37]. Low-start fast responders initially demonstrated the highest rate of recovery, with their trajectory encompassing all the dynamic range of the FIM scale, and quickly detaching from slow responders’ one. However, top- and bottom-scale saturations affect the recovery rates of high-start fast responders and non-responders in this study. Assessment through scales for disorder of consciousness should be applied to the latter group [38].

Long-term decline in function in the years following ABI is a matter of concern. Adult studies report inconsistent findings [1,2,39], with no [2] or little evidence [39] of functional decline starting 5 years post-injury. Hammond et al. [2] found that the proportion of adults with DoC who achieve functional independence increases between 5 and 10 years post-injury, and especially among those who show late command-following. Conversely, Corrigan et al. [39] found that 39% of TBI survivors in USA deteriorated from a global outcome attained 1 or 2 years postinjury, regardless of age. We found no evidence of long-term trajectory deterioration for three groups out of four (see Figure 1a right panel and Figure 1b). However, enlarged confidence intervals cannot exclude it for the high-start fast responders, due to the scant long-term data in this class, and to the ascertained negative bias towards complex cases over time. Inspection of individual trajectories revealed only 8/92 instances of decline in this group, starting, on average, at year 4. Further research needs to clarify the risk of deterioration in high functioning groups, as this is relevant to life satisfaction [40,41].

Based on data at first discharge, we were able to predict whether a child would match the favorable recovery curve (responders vs. non-responders) over the long term with very high accuracy (0.93). Prediction among all the four trajectories remained high (0.80), with very good prediction for the non-responders (0.88), and errors committed only between neighboring trajectories. The feature most contributing to prediction was the FIM difference between first admission and discharge. This confirms that, by the end of the in-stay, the type of rehabilitation trajectory—but not the outcome—has already been established on average. Of note, when considering single FIM domains, the subjects who respond to treatment slowly, and particularly in selfcare and cognition, are the hardest to predict.

Our method enables us to draw predictions of the long-term functional course and outcome of one single patient for a given rehabilitation program, at first discharge. Early identification of non-responders enhances: (1) timely allocation of different available therapeutic resources or strategies to the targeted patient, (2) further investigation of unmet needs, (3) optimization of the patient’s therapeutic involvement, and (4) savings on ineffective treatments.

### Study Limitations

This is a single-center study. The prediction method was internally validated, but it needs external validation in a second center and with different assessment tools. Internal validation is sufficient for the creation of tools to be deployed in single centers, yet major changes in the institutional rehabilitation programs over time can deteriorate or invalidate their effectiveness in practice.

FIM/WeeFIM demonstrated both flooring and ceiling saturation despite normalization; negative bias in functional performance over time was described but not corrected; and some data were missing not at random at the latest timepoints.

The amount of intensive treatment delivered in-hospital was assumed proportional to LOS; despite the fact that this is generally true for our cohort, the type of intervention (e.g., physiotherapy, speech therapy, etc.) was not fine-grained quantified. Home-based rehabilitation following discharge was standardized, but our center did not have full control over the delivery. This might have injected variability in the recovery trajectories and outcome. LOS was different in patients, and dependent on both the overall patients’ clinical condition and family compliance. Additionally, similarly to most clinical studies, the predictive variables were not completely independent; GCS, time from event to admission, GOS at admission, motor damage and presence of spasticity were not included to avoid statistical dispersion, due to collinearity. Hence, their effects cannot be separated from those produced by the variables considered. Socioeconomic factors were not considered either, having resulted negligible in the context where data were acquired [42].

We aimed at the accurate prediction of the recovery trajectories, based on ideally all the information available at first discharge. Early outcome prediction in the acute ABI [43] (in the ICU or at admission) is another very urgent need, yet requiring different methodological choices.

## 5. Conclusions

The patients’ heterogeneity of response to specific rehabilitation treatments is a key confound to medical decisions and outcome prediction, as study cohorts typically include responders on distinct recovery trajectories. We found subgroups of responders in a cohort of 600 pediatric patients with ABI caused by mixed etiologies. We uncovered four data-driven subtypes of trajectories based on distinct temporal progression patterns, and we predicted treatment response at single-subject level based on the clinical and functional condition at first discharge. The method enables a conceptual shift from outcome-based patient classification to stratification of the temporal evolution of the response to a specific treatment program. Our results challenge the statement that “slow recoveries tend to be poor recoveries” [19] in children, and identify a group of slow responders, who fail to reach good functional level by year 1, but continuously improve until approaching very good functional recovery by year 7. We wish that our research raises awareness in professionals that individuals at risk of slow recovery need to be identified, as they can conceal great potential. We need joint efforts to develop sustainable intervention strategies for optimizing these patients’ involvement over prolonged times.

## Figures and Tables

**Figure 1 jpm-11-00675-f001:**
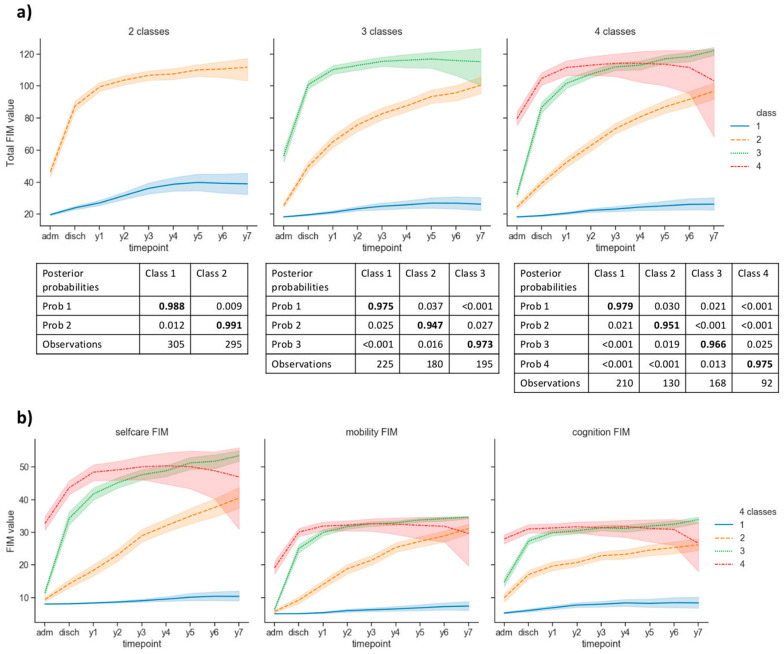
**(a)** Typological analysis. From left to right: two, three and four classes clustering. Two-classes clustering separated what we called responders (orange) from non-responders (blue). Three classes clustering isolated fast- (green), slow- (orange) and non-responders (blue). Four classes clustering separated high-start fast (red), low-start fast (green), slow (orange) and non-responders (blue). “adm” and “disch” indicate first admission and discharge, respectively. Numbers in bold indicate the posterior probabilities for each trajectory to be assigned to the correct class. **(b)** From left to right, the same typological analysis identified four classes of trajectories in each FIM domain: selfcare, mobility and cognition.

**Figure 2 jpm-11-00675-f002:**
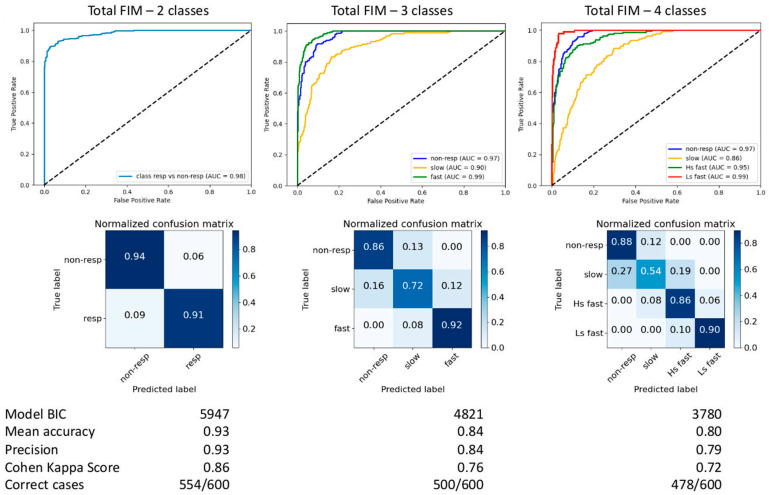
From left to right, prediction performance on previously unseen cases, among two, three and four classes of trajectories. Performance is indicated as per-class area under the curve (AUC in legends), and as overall classification accuracy, precision, Cohen kappa score on the test set, and number of correctly predicted cases. Confusion matrices for two, three and four classes predictions show normalized successful predictions along the main diagonal, and prediction errors off-diagonal. Errors were committed between neighboring trajectories in all cases.

**Figure 3 jpm-11-00675-f003:**
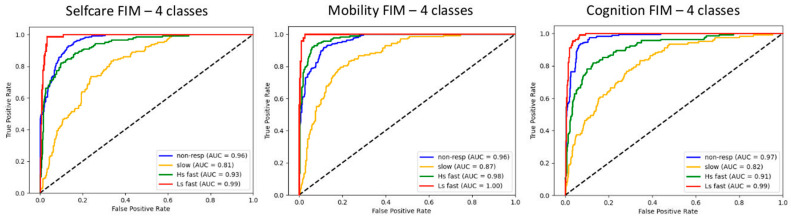
From left to right, prediction performance on previously unseen cases, among four classes of trajectories in each FIM domain: selfcare, mobility and cognition. “non-resp” is non responders; “slow” is slow responders; “resp” is responders; “fast” is fast responders; “Hs” is high-start fast responders; and “Ls” is low-start fast responders.

**Table 1 jpm-11-00675-t001:** Clinical and demographic characteristics of the total sample and the two groups divided by traumatic/non-traumatic etiology. Description includes the symptoms associated with the primary motor disorder and the need for decompressive craniotomy/neurosurgery (DC/N) in acute.

	Total Sample *N* = 600	Traumatic *N* = 276	Non-Traumatic *N* = 324
Age at event (months)	Mean (SD)	Mean (SD)	Mean (SD)
	89.5 (62.1)	106.6 (63.0)	74.9 (57.4)
	Median [IQR]	Median [IQR]	Median [IQR]
Days of coma *	20 [7; 80]	20 [10; 67.5]	15 [7.5; 90]
Time from event to admission (months)	43 [27; 69]	39.5 [26; 61]	45 [28; 73]
	*N* (%)	*N* (%)	*N* (%)
Patients in unresponsive wakefulness syndrome for at least one year	81 (13.5)	27 (9.8)	54 (16.7)
	Median (Mode)	Median (Mode)	Median (Mode)
GCS score at event	6 (3)	5 (3)	6 (3)
GOS score at admission	3 (3)	3 (3)	3 (3)
DRS at admission	20 (24)	19 (22)	20 (24)
GOS score at discharge	3 (3)	4 (3)	3 (3)
DRS at discharge	8 (5)	7 (5)	11 (5)
	*N* (%)	*N* (%)	*N* (%)
Male	367 (61.2)	187 (67.8)	180 (55.6)
Female	233 (38.8)	89 (32.2)	144 (44.4)
	*N* (%)	*N* (%)	*N* (%)
Cranial fracture at event	120 (20.0)	120 (43.5)	0 (0.0)
DC/N in acute	288 (48.0)	153 (55.4)	135 (41.7)
Epilepsy during in-stay	163 (27.2)	55 (19.9)	108 (33.3)
Motor impairment ad admission:	*N* (%)	*N* (%)	*N* (%)
Quadriparesis	322 (53.7)	139 (50.4)	183 (56.5)
Right hemiparesis	85 (14.2)	36 (13.0)	49 (15.1)
Left hemiparesis	78 (13.0)	40 (14.5)	38 (11.7)
Paraparesis	7 (1.2)	3 (1.1)	4 (1.2)
Ataxia	40 (6.7)	18 (6.5)	22 (6.8)
Minimal dysfunction ^$^	40 (6.7)	23 (8.3)	17 (5.2)
None	28 (4.7)	17 (6.2)	11 (3.4)

* Patients who remained in unresponsive wakefulness syndrome (i.e., persistent vegetative state) after three months from event are omitted from the count of days of coma, according to the current definitions. ^$^ Minimal dysfunction was diagnosed based on the following criteria: (1) functional or neurologic examination not completely within the normative ranges for age, and (2) presence of mild motor impediment (e.g., difficulties in fine movements, mild motor coordination deficit, lag or slowdown in motor execution), with no overt pyramidal or extra-pyramidal signs. GCS is Glasgow Coma Score; GOS is Glasgow Outcome Score; DRS is Disability Rating Scale; DC/N is decompressive craniotomy/neurosurgery.

**Table 2 jpm-11-00675-t002:** Descriptive statistics, including cumulative missing FIM assessments over data points, and divided per deaths, exit from service due to recovery (medical decision), non-recommended opt out, and future data points (censored).

	Admission	Discharge	Year 1	Year 2	Year 3	Year 4	Year 5	Year 6	Year 7
FIM domains	Median [IQR]	Median [IQR]	Median [IQR]	Median [IQR]	Median [IQR]	Median [IQR]	Median [IQR]	Median [IQR]	Median [IQR]
Total	18 [18; 38]	38 [18; 85]	48 [19; 100]	60 [21; 105]	67 [20; 104]	69 [20; 105]	79 [19; 112]	69 [18; 106]	51 [18; 105]
Selfcare	8 [8; 12]	11 [8; 33]	16 [8; 42]	23 [8; 42]	25 [8; 43]	27 [8; 45]	30 [8; 48]	25 [8; 45]	18 [8; 45]
Mobility	5 [5; 6]	8 [5; 27]	14 [5; 31]	19 [5; 32]	20 [5; 32]	24 [5; 32]	26 [5; 33]	22 [5; 33]	17 [5; 34]
Cognition	5 [5; 17]	15 [5; 27]	17 [6; 29]	18 [7; 30]	20 [7; 30]	20 [7; 30]	22 [6; 30]	19 [5; 30]	16 [5; 28]
	*N* (%)	*N* (%)	*N* (%)	*N* (%)	*N* (%)	*N* (%)	*N* (%)	*N* (%)	*N* (%)
Missing data points	0 (0)	0 (0)	92 (15)	183 (31)	278 (46)	350 (58)	407 (68)	462 (77)	497 (83)
Deaths *	-	-	4 (1)	7 (1)	10 (2)	14 (2)	14 (2)	23 (4)	23 (4)
Recovery	-	-	31 (5)	62 (10)	81 (14)	92 (15)	100 (17)	107 (18)	110 (18)
Opt out	-	-	49 (8)	89 (15)	120 (20)	144 (24)	152 (25)	158 (26)	159 (27)
Censored	-	-	8 (1)	25 (4)	67 (11)	100 (17)	141 (24)	174 (29)	205 (34)

* Deaths were computed for the patients enrolled in the study and admitted to the post-acute rehabilitation center. Thus, they do not include patients who died during intensive care, before potential referral. FIM is Functional Independence Measure; IQR is interquartile range [1st–3rd].

## Data Availability

A subsample of this cohort was previously investigated. The complete dataset is available at https://zenodo.org, accessed on 17 July 2021, doi:10.5281/zenodo.4153962, uploaded on 29 October 2020.

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
