# Peer review of "Individualized Prognostic Prediction of the Long-Term Functional Trajectory in Pediatric Acquired Brain Injury"

_jpm, 2021, doi:10.3390/jpm11070675_

Round 1

Reviewer 1 Report

The authors present work and conclusions on acquired brain injury in children and a possible process to detect slow improvement in FIM early on to explore alternative therapeutic and rehabilitative regimens to improve prognosis for these patients. The authors present, clearly and cohesively, all their results and draw reasonable conclusions from the years of work they performed. 

This work can be instrumental in understanding progress patterns in children suffering from TBI and may have implications for adult TBI as well. 

Author Response

9 July 2021

Dear Reviewer 1,

With reference to your correspondence of 22 June 2021, we thank you for the quick feedback, and we are pleased to hear that our manuscript was well-received and considered for publication in The Journal of Personalized Medicine.

Please find attached a revised version of our manuscript, with tracked changes.  In particular, we would highlight that we had the manuscript reviewed for the English language and grammar.

We detail below our response.

We are grateful for the fast turnaround, and we hope that the additional changes made at this stage are satisfactory.

We remain available for clarification and in case additional adjustments are required, and we look forward to hearing from the Editorial Office.

Yours sincerely

Erika Molteni and Elena Beretta, on behalf of all the authors

" The authors present work and conclusions on acquired brain injury in children and a possible process to detect slow improvement in FIM early on to explore alternative therapeutic and rehabilitative regimens to improve prognosis for these patients. The authors present, clearly and cohesively, all their results and draw reasonable conclusions from the years of work they performed. 

This work can be instrumental in understanding progress patterns in children suffering from TBI and may have implications for adult TBI as well. "

-> We thank Reviewer 1 for the positive feedback on the manuscript.

We would like to highlight that the manuscript was reviewed by an English-native speaker for grammar and spelling.

Reviewer 2 Report

I think it is an interesting paper and that it addresses a topic with little
scientific literature, however, I believe that there are several
limitations in this study, starting with the tool used and the non-inclusion
or measurement of the treatments received, if these objectives are intended
this should to be included

Lines 41-45.- The authors refer to the FIM as the most widely used functional tool, but the reference used by Granger is from the year 1985. The authors must include a greater number of references with a high level of evidence that supports this statement of the use of this tool and not another at a functional level.

Lines 54-56.- authors should review this paragraph, wording is unclear

Lines  57-58.- It should expand and explain those other tools

Table 1.- Hemiparesis in two lines and can you explain what they refer to with minimal dysfunction and how was it evaluated? It is not clear to me..

Table 2.- As the data in this table is exposed, the interquartiles are confusing for me and why do they not include the deviation?

Lines 199- 295 - Section 3.3. It is tedious to read, they duplicate information that is seen in Figure 1. Being the thickness of the study, some way to expose the data more visually should be considered.

Figure 1.- it does not have good resolution, nor size. As I said, it is tedious to read

Lines 226-227.- I do not understand why they do not explain these domains or their comparisons and then include it in figure 1 b

Figure 2.- The authors introduce excessive information in the figures that hardly the legend is perceived

Line 291.- it is a pity not to have these data in a longitudinal study and that would enrich the study, in addition I believe that not having data or not showing them of the intervention that has been carried out with these children is a loss of outcomes in such a large sample, although the authors mention studies in adults do not include in children this should be modified

Line 298- The authors should expand and structure these lines.

Line 300 The authors say they have no evidence of impairment but the tools used, although pertinent, are insufficient, as it affects occupational performance and the differences in these four groups should be discussed or put as a major limitation of the study. The FIM is really valid but it has few domains and the cognitive one is quite scarce.

Line 316.- Creo que esto debe ser más ampliamente discutido están hablando de una actividad de la vida diaria esencial, ya que los autores decidieron usar esta herramienta y no otra esto debería ser discutido. Por ejemplo porque no se valoró a nivel de integración sensorial? O porque no se valoró con el AMPS....

Line 332.- This is a major limitation of the present study, neither is it quantified nor is almost mentioned, neither the intensity nor the type of intervention, such as the approach and approaches of the intervention team ... were there no occupational therapists? It surprises me that they only talk about physiotherapists and speech therapists. In addition, the authors do not refer to the children's environments, including the socioeconomic profile, or even if they are children from rural or urban areas ... what professionals do they turn to ... this has repercussions and they are essential factors such as the school environment ...

The authors aimed to design a framework for single-subject prediction of trajectories, based on the clinical and functional conditions at first discharge. Without this information, how can they meet this objective?

Line 361.- The authors aimed to design a framework for single-subject prediction of trajectories, based on the clinical and functional conditions at first discharge. Without this information, how can they meet this objective?

References should be reviewed, many do not appear in the correct format, they do not include doi ..

Author Response

9 July 2021

Dear Reviewer 2,

With reference to your correspondence of 2 July 2021, we thank you for the quick feedback, and we are pleased to hear that our manuscript was considered for publication in The Journal of Personalized Medicine.

Please find attached a revised version of our manuscript, with tracked changes. In particular, we would highlight that we have extended some paragraphs, according to what was highlighted in Your review.

In addition, we detail in the word file attached below a point-by-point response to Your comments.

We are grateful for the fast and constructive turnaround, and we hope that the additional changes made at this stage are satisfactory.

We remain available for clarification and in case additional adjustments are required, and we look forward to hearing from the Editorial Office.

Yours sincerely

Erika Molteni and Elena Beretta, on behalf of all the authors

Reviewer 2

  • I think it is an interesting paper and that it addresses a topic with little
    scientific literature, however, I believe that there are several 
    limitations in this study, starting with the tool used and the non-inclusion
    or measurement of the treatments received, if these objectives are intended 
    this should to be included

We thank the reviewer for their appreciation of the manuscript. We address here below the limitations:

  • The tool used: We agree with the reviewer that, although Functional Independence Measure, and its pediatric version WeeFIM are two scales for the assessment of the motor, cognitive and self-care functions widely accepted worldwide, they may not be employed in some centres, and an absolute standard is still missing in this field. However, there are recommendations encouraging the use of this tool in the functional assessment of acquired brain injury (Ardolino et al., 2012), and evidence of FIM superiority to some other instruments, such as PEDI (Williams et al., 2017) for the assessment in pediatrics. Of note, if any other better tool exists for the functional assessment in pediatrics, or will be designed in the future, arguably it will lead to more decisive findings and more precise predictions than the ones presented in the manuscript. In this perspective, the employment of FIM can be seen as a minimum standard through which good rehabilitation trajectory prediction can be achieved.
  • Non-inclusion/measurement of the treatments received: The intervention is described in the supplementary material 4, and details are reported for both the in-staying period and home-based treatment. However, we acknowledge that children remained in the rehabilitation service for a variable time duration, which was also dependent on the child’s initial condition and functional improvement during recovery. We also acknowledge that the number and intensity of treatments received were not included in the model, as they were not considered subject-dependent.
  • Lines 41-45.- The authors refer to the FIM as the most widely used functional tool, but the reference used by Granger is from the year 1985. The authors must include a greater number of references with a high level of evidence that supports this statement of the use of this tool and not another at a functional level.

We thank the reviewer for pointing out this weakness. The paragraph has been extended as follows. Changes are tracked in blue here

Response is generally assessed through scales, of which the Functional Independence Measure (FIM) (Granger et al., 1985) is the one recommended and most frequently used (Ardolino et al, 2012).

FIM was also employed in public health, to determine the prevalence and nature of residual disability after inpatient rehabilitation for children with traumatic injuries (Zonfrillo et al., 2013). In a recent systematic review studying children with acquired brain injury (Williams et al., 2017), the paediatric version of the FIM scale (WeeFIM) resulted the elective tool for functional assessment, being more suitable for the inpatient setting, and quicker to administer, compared to the Pediatric Evaluation of Disability Inventory (PEDI). It also showed less ceiling effects. WeeFIM was also employed in paediatric cohorts with acquired brain injury as a measure of global functional outcome at admission (Marino et al., 2018) and discharge from hospital (Suskauer et al., 2009; Shaklai et al., 2014; Maddux et al., 2018), and as a robust clinical indicator of motor function correlating with neuroimaging biomarkers, such as the DTI measures in the corticospinal tract at DTI (Ressel et al., 2018; Ressel et al., 2017) and the volume of putamen (Molteni et al., 2019). However, although FIM can track functional improvements reliably over time, there is no score cut-off to predict subsequent recovery trajectory or outcome.

  • Lines 54-56.- authors should review this paragraph, wording is unclear

We thank the reviewer for pointing out this weakness. The sentence was clarified as follows (changes are tracked in blue here):

Similarly to observations in adults, pediatric studies described subgroups who did not show any recovery trend. Limitations were short timeframe of observation [19] and employment of functional assessments other than FIM, such as the Gross Motor Function Measure-66  [22].

  • Lines  57-58.- It should expand and explain those other tools

The sentence was expanded as follows (changes are tracked in blue here):

Limitations were short timeframe of observation [19] and employment of functional assessments other than FIM, such as the Gross Motor Function Measure-66 [22].

Unfortunately, we cannot provide more examples here, as we could not find any additional supporting literature. Should the reviewer be aware of any piece of literature we did not consider, we certainly remain available to make further additions to the manuscript based on their indications.

  • Table 1.- Hemiparesis in two lines and can you explain what they refer to with minimal dysfunction and how was it evaluated? It is not clear to me.

Right and left hemiparesis are now reported in two distinct lines in Table 1, as requested.

Minimal dysfunction was diagnosed based on the following criteria:

- functional or neurologic examination not completely within the normative ranges for age;

- presence of mild motor impediment (e.g., difficulties in fine movements, mild motor coordination deficit, lag or slowdown in motor execution);

- no overt pyramidal or extra-pyramidal signs.

After the reviewer’s remark, these criteria are now reported as a footnote of Table 1 in the main text of the manuscript.

  • Table 2.- As the data in this table is exposed, the interquartiles are confusing for me and why do they not include the deviation?

In descriptive statistics, measures which distribute with Gaussian probability, also termed “normal”, can be summarised through mean and variance (or standard deviation). Measures which distribution does not follow a Gaussian probability curve, or for which Gaussianity cannot be inferred, are usually described through median (i.e., quantity lying at the midpoint of the frequency distribution of observed values, such that there is an equal probability of falling above or below it) and inter-quartile range (IQR, i.e., statistical dispersion to 25% and 75% of the whole rank-ordered data set, divided into four equal parts).

As FIM is considered a categorical measure in this context, and FIM values clearly distribute in a non-Gaussian (i.e. non-normal) fashion, the use of mean and standard deviation is not appropriate for providing statistical description. Medians and IQRs are appropriate descriptors instead.

  • Lines 199- 295 - Section 3.3. It is tedious to read, they duplicate information that is seen in Figure 1. Being the thickness of the study, some way to expose the data more visually should be considered.

We thank the reviewer for this question, and we include here below a more graphical representation of the results listed in the section. We hope this helps to clarify. However, we would like to point out that this two-dimensional form of representing the main results does not fully present all the significant relations between classes for each covariate and for each clustering (with 2, 3 and 4 classes). For this reason, it is less rigorous than the full list of significant tests reported in section 3.3. However, we remain available to add the image in the main manuscript in case the reviewer deems it is needed.

  • Figure 1.- it does not have good resolution, nor size. As I said, it is tedious to read

Figures 1 and 2 in the main manuscript were enlarged in size. However, the images will be provided in high resolution to the Editorial office of the journal, who will arrange the best resolution, size and location in the manuscript. We remain available to the Editorial office to assist with the graphical details of these images.

  • Lines 226-227.- I do not understand why they do not explain these domains or their comparisons and then include it in figure 1 b

We thank the reviewer for this question, and we acknowledge this is one of the most subtle detail of the manuscript, which could be misleading for the reader. The “results not shown” mentioned at former lines 226-227 (now lines 283-284) are not those depicted in figure 1b, although extremely similar.

In the manuscript, we describe the clustering of sub-groups and corresponding rehabilitation trajectories, based on the total FIM scores. This means that:

- clustering was conducted on the longitudinal total FIM values and covariates.

- then, based on the sub-groups identified by clustering on total FIM and covariates, we could plot the total FIM trajectories (figure 1a), and the trajectories of each FIM domain (figure 1b), namely selfcare, mobility and cognition.

However, clustering can be conducted on one FIM domain only.

What is “not shown” is the result of the following additional processing:

- clustering was conducted on the longitudinal selfcare FIM values and covariates.

- then, based on the sub-groups identified by clustering on selfcare FIM and covariates, we could plot the total FIM trajectories, and the trajectories of each FIM domain: selfcare, mobility and cognition.

- clustering was then conducted on the longitudinal mobility FIM values and covariates.

- then, based on the sub-groups identified by clustering on mobility FIM and covariates, we could plot the total FIM trajectories, and the trajectories of each FIM domain: selfcare, mobility and cognition.

- clustering was then conducted on the longitudinal cognition FIM values and covariates.

- then, based on the sub-groups identified by clustering on cognition FIM and covariates, we could plot the total FIM trajectories, and the trajectories of each FIM domain: selfcare, mobility and cognition.

This provides other 3x6=18 panels depicting the trajectories identified by clustering subjects based on these different FIM domains. Results are largely equivalent.

What matters here is indeed the (almost) invariance of results, given different clustering criteria, i.e. based on different FIM domains.

The overall conclusion is that, regardless of whether the clustering was conducted on total FIM scores or on scores in one of its domains (selfcare, mobility or cognition), subjects are grouped very similarly, and this results in very similar group trajectories. Clinically, this implies that the assignment of a patient to their class is independent from the specific methodological choices operated to perform the clustering, and is stable across the FIM domains.

As a further detail, we point out that clustering on the longitudinal selfcare FIM values and covariates provided slightly larger confidence intervals and lower accuracy in delineation of the trajectories. This finds probable clinical interpretation in the empirical observation that selfcare abilities are usually (re)gained later in the rehabilitation path, compared to basic motor and cognitive abilities measured by the other two FIM domains, and by a subset of patients only. However, the reader might argue that clustering based on longitudinal selfcare FIM only is probably the least suitable approach for cohorts of such average severity. Consequently, we tend to consider these additional analyses as confirmatory of the clustering conduced on the total FIM scores, rather than being a core methodological part. For this reason, data were not shown.

We added the following text into the discussion, at lines 335-339 of the manuscript:

Regardless of whether clustering was conducted on total FIM scores or on scores at specific FIM domains (selfcare, mobility or cognition), subjects grouped consistently, and produced similar group trajectories. Clinically, this implies that the assignment of a patient to their group remains fairly independent from the specific clustering method, and is stable across the FIM domains.

  • Figure 2.- The authors introduce excessive information in the figures that hardly the legend is perceived

Figure 2 has now been subdivided into Figure 2 (former figure 2a) and Figure 3 (former figure 2b). The images will be provided in high resolution to the Editorial office of the journal, who will arrange the best resolution, size and location in the manuscript. We remain available to the Editorial office to assist with the graphical details of these images.

  • Line 291.- it is a pity not to have these data in a longitudinal study and that would enrich the study, in addition I believe that not having data or not showing them of the intervention that has been carried out with these children is a loss of outcomes in such a large sample, although the authors mention studies in adults do not include in children this should be modified

We confirm to the reviewer that the missing data we described in the manuscript cannot be obtained unfortunately, as patients dropped out from the rehabilitation service as described.

The intervention is described in the supplementary material 4, and details are reported for both the in-staying period and home-based treatment. However, we acknowledge that children remained in the rehabilitation service for a variable time duration, which was also dependent on the child’s initial condition and functional improvement during recovery. We also acknowledge that the number and intensity of treatments received were not included in the model, as they were not considered subject-dependent.

  • Line 298- The authors should expand and structure these lines.

The sentence was expanded as follows (changes are tracked in blue here):

Long-term decline in function in the years following ABI is a matter of concern. Adult studies report inconsistent findings [1,2,39], with no [2] or little evidence [39] of functional decline starting 5 years post-injury. Hammond et al. (Hammond et al., 2019) found that the proportion of adults with DoC who achieve functional independence increases between 5 and 10 years post-injury, and especially among those who show late command-following. Conversely, Corrigan et al. (Corrigan et al., 2014) found that 39% of TBI survivors in USA deteriorated from a global outcome attained 1 or 2 years postinjury, regardless of age.

  • Line 300 The authors say they have no evidence of impairment but the tools used, although pertinent, are insufficient, as it affects occupational performance and the differences in these four groups should be discussed or put as a major limitation of the study. The FIM is really valid but it has few domains and the cognitive one is quite scarce.

We clarified this point by adding the words “long-term trajectory” to the sentence in the main text of the manuscript, which now reads as follows:

We found no evidence of long-term trajectory deterioration for 3 groups out of 4.

Here we are discussing that fact that, based on the confidence intervals (shown in fig 1), we can exclude deterioration of the recovery trajectory in the long term for classes 1, 2 and 3, but not for class 4 (red line, high-start fast responders class), based on FIM assessments. The (still possible) deterioration is discussed in terms of a long-term slight decline of the trajectory of class 4 assessed through the FIM scale, and by no means we are implying or suggesting any presence/absence of evidence of impairment otherwise defined. In addition, in the same paragraph we warn the reader about two factors which might have influenced the trajectory of class 4: negative bias due to drop-out from the service and scantiness of data over the long-term for this class. So, all considered, the data should be interpreted with opposite approach: if for classes 1, 2 and 3 we can statistically rule out overall long-term decline in FIM scores, this cannot be stated for class 4, i.e., for the high-start fast responders class), for whom the presence/absence of FIM scores decline remains uncertain.

  • Line 316.- Creo que esto debe ser más ampliamente discutido están hablando de una actividad de la vida diaria esencial, ya que los autores decidieron usar esta herramienta y no otra esto debería ser discutido. Por ejemplo porque no se valoró a nivel de integración sensorial? O porque no se valoró con el AMPS....

The Assessment of Motor and Process Skills (AMPS) is currently used at IRCCS E. Medea, where the cohort received rehabilitation and was studied. The centre is also a “Continuing Professional Education” national site for the training of professionals who need to learn how to apply AMPS. However, AMPS was not used for the description of the cohort described in this manuscript. AMPS evaluates the performance of activities of daily living that the subject choses (or the therapist choses, in agreement with the child and/or caregivers, depending on age). (Some) goals are selected based on the relevance and significance for the subject. This freedom of choice introduces a degree of subjectivity in the assessed abilities, which would need to be statistically handled, and which is beyond the scope of this work. For this reason, we employed a totally standardized quantitative tool (FIM), which was identically applied in the assessment of all patients, regardless of their perception of the relevance of single activities and abilities. This was done to increase the rigour of the study design and to simplify the statistical analyses.

  • Line 332.- This is a major limitation of the present study, neither is it quantified nor is almost mentioned, neither the intensity nor the type of intervention, such as the approach and approaches of the intervention team ... were there no occupational therapists? It surprises me that they only talk about physiotherapists and speech therapists. In addition, the authors do not refer to the children's environments, including the socioeconomic profile, or even if they are children from rural or urban areas ... what professionals do they turn to ... this has repercussions and they are essential factors such as the school environment ...

Treatment. As mentioned above, the intervention is described in the supplementary material 4, and described for both the in-staying period and home-based treatment. Length of staying in the centre, however, was variable for children, depending on the children’s initial severity, needs and progress during the rehabilitation course. Number and intensity of treatments were not included in the model, as they were not considered subject-dependent; however we acknowledge this might be seen as a limitation of the study, which we disclose in the “Limitations” section of the manuscript.

Role of the occupational therapist. In Italy the professional role of the occupational therapist (OT) is fully recognised. However, in Italian rehabilitation hospitals the presence of OTs is normally lower, if compared with some other areas of the world (e.g., South America). At IRCCS E. Medea, where our patients were admitted, treated, and followed up, OTs work within the physiotherapy service, and do not operate in an independent OT service. However, although working in such shared environment, OTs are specifically in charge of setting goals for the daily life independence and working with our children to maximise their capabilities of independent living. They also work conjointly with the “orthotics and assistive devices” unit at our centre. For these reasons, occupational therapies do not appear as treatments independent from physiotherapy sessions during in-hospital staying. Conversely, in local services, OT sessions are distinct provision with respect to physiotherapy treatments, as indeed specified in the Supplementary Material 4, at the paragraph “standardized home-based treatment”. These aspects are mainly related to the organization of the different national healthcare systems and national public health policies in place, and for this reason we did not give much emphasis in the main manuscript. Of note, all the children enrolled in our cohort were served by the same public (Italian) national healthcare system, and were thus subject to the same pathways of care.

Socioeconomic profile. All the children enrolled in our cohort were assisted in the same centre, and through the same public (Italian) national healthcare system. Thus, all children were subject to the same pathways of care. However, the home-based therapy was delivered by different local services, although all operating within the same national healthcare system. Of note, the Italian national healthcare system is universal and free over all the country. All considered, we deem that the socioeconomic profile impacted marginally on the chances of hospitalization, provision of care, and home-based treatment for the children enrolled. As a secondary matter, though, the socioeconomic status of the family could have impacted on chances of additional private home-care and care seeking behaviors, chances of better mental health support for the caregivers, and better facilities for daily living (e.g., better nutrition, larger spaces at home, cleaner environment at home, etc). Although we do not have availability of all this information for our cohort, previous work conducted by our group focused on the quantification of the socioeconomic correlates of care and recovery in similar patients, accessing the same center (DOI: 10.1177/0883073813513329, end of page 6). One sentence was added at current line 436 of the manuscript to specify this:

Socioeconomic factors were not considered either, having resulted negligible in the context where data were acquired [42].

Geographical factors. Italy is a small country, with a very good ambulance service coverage nationwide, including helicopter rescue, and excellent public hospital care. Safety is good on the whole road system, including secondary roads, and traumatic brain injuries have been decreasing steadily over the last 20 years. This is partly counterbalanced by a mildly increasing trend of encephalitis cases, including severe presentations, despite absence of relevant endemic diseases causing epidemics (e.g., malaria etc.). In this context it is difficult to draw a separation between urban and rural areas in terms of healthcare provision. The majority of children living in small towns have access to local A&Es and/or hospitals, and can be transferred to tertiary centres in the main cities with ease and through routinary short transfers, if not directly admitted to tertiary centres when severity of disease presentation requires so. For these reasons we did not consider any sub-grouping according to urban/rural areas in our analyses.

Additional professionals who take patients in charge. As mentioned above, the intervention is described in the supplementary material 4, where we report details for the home-based treatment. Outpatient rehabilitation services included physical, occupational, and speech therapy. Follow-up assessments were preformed yearly by rehabilitation physicians at our center. Families/caregivers were involved in the interventions, and they were trained to act in first person to intervene for improving attention, memory, executive functioning, and emotional/behavioral functioning in daily life and home setting.

School environment. All patients in this cohort returned home after first discharge from hospital, with the exception of 3 very severe subjects who needed a 24-h nursing presence, and were admitted in care homes. These 3 patients never went back to school due to the severity of their condition. Only ~50% of all the other patients in vegetative state/unresponsive wakefulness syndrome could go back to some form of schooling. Patients who did not go back to any form of schooling were prevalently dependent on some device (eg. ventilators, feeding devices, etc.). All non-VS patients could go back to some form of schooling, including special education. All children with IQ<70 have the right to access special teaching, to receive a personalized learning plan, and to be taught by a dedicated teacher with special training on disabilities in Italy. Children who have IQ>70 but a demonstrated specific deficit (e.g., memory or executive functions deficit) usually receive a personalized learning plan, and some hours of teaching by a dedicated teacher with special training on disabilities. During hospital in-stay, all children in our cohort attended our hospital school, unless being in a coma or vegetative state. As a standard procedure, for each child who, once discharged from our hospital, can attend some form of schooling, a pedagogist from our centre gets in contact with the corresponding local school. The local school identifies the reference teacher for the child, who makes arrangements with our pedagogist before the child returns to school. During these contacts, special arrangements are usually agreed and made, both in terms of facilities/ergonomic solutions in the class and teaching goals/contents/adaptations. This schooling re-integration model was presented at international conferences (e.g. IpBIS, Rome 2017), but never published on peer-reviewed journals.

  • Line 361.- The authors aimed to design a framework for single-subject prediction of trajectories, based on the clinical and functional conditions at first discharge. Without this information, how can they meet this objective?

Yes, we confirm that the one stated by the reviewer is the main aim of this work.

At first discharge from hospital, the following information was available for all the subjects: FIM scores at admission and discharge, age at event, days of coma (truncated at the length of stay in the hospital, if longer), length of stay in the hospital, etiology, need of decompressive surgery, and insurgence of epilepsy. This information entered the predictive model for unseen cases, and the rate of right/wrong prediction on the single unseen cases is reported in the manuscript, depending on the type of prediction (e.g., responder/non responder, or more fine grained prediction with three or four classes).

If the perplexity of the reviewer is about the robustness of the results reported in the manuscript, we remain available for further clarifications. Also, we point out here that the data are freely available, and the method is described in such a way that can be fully replicated on new data.

If the perplexity of the reviewer is about the availability of these same measures in prospective cases and other centres, we wish to specify that the methodological framework can be applied to longitudinal trajectories acquired through any functional assessment tool (i.e. different from FIM), and that the clinical data are very basic measures, most likely acquired in any hospital at discharge, or collectable from caregivers. Clearly, the prediction precision, and more generally the method performance, will depend on the functional assessment tool used and the number of patients employed for training the algorithm.

  • References should be reviewed, many do not appear in the correct format, they do not include doi ..

We thank the reviewer for having pointed out this matter. We reviewed the references and added the missing DOIs. Of note, a couple of documents do not have doi (one very old article and one technical document). For these, “doi unavailable” was explicitly indicated in the corresponding reference.